# On the Generalization Capabilities, Design Choices and Limitations of Keypoint Imitation Learning

Anonymous* - Double Blind Submission

*Abstract*—RGB-based imitation learning requires many demonstrations to generalize to unseen objects or scenes, motivating research into intermediate representations to improve generalization for robotic manipulation. Vision foundation models enable one-shot extraction of keypoints to provide such a representation. However, how to optimally integrate keypoints into imitation learning and when they outperform alternative representations remains unclear. We systematically study design choices in keypoint imitation learning (KIL), thereby consolidating insights from prior work into practical guidelines. Evaluating over 2000 real-world rollouts across five tasks and diverse scene variations, KIL achieves a 75% overall success rate, substantially outperforming an RGB baseline (47%) and performing similar to $S^2$-diffusion (73%), an object-centric baseline. Finally, we explore the limitations of the foundation models used for keypoint extraction and find that they are sensitive to large variations in object orientation.

Our results confirm KIL as a data-efficient approach for robot learning, and suggest directions for future research to improve our understanding of its limitations and potential.

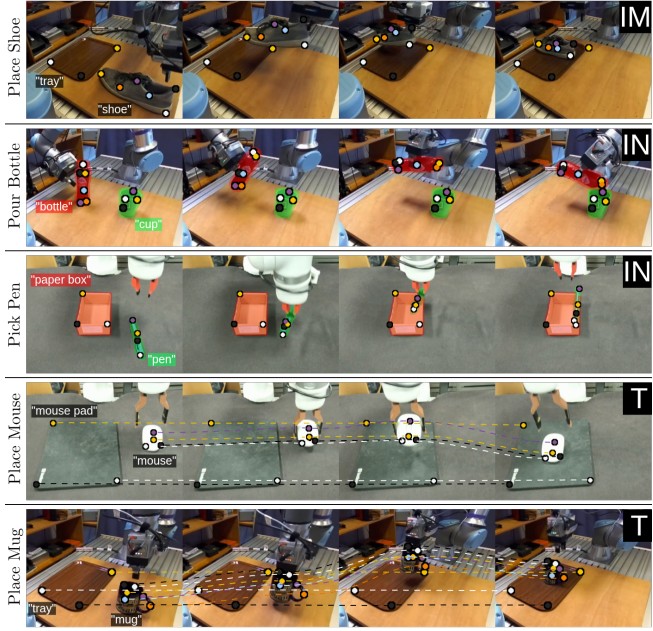

Fig. 1. Successful real-world rollouts of our keypoint imitation learning policies. The top-right label indicates the keypoint extraction method used: IM=image matching, IN=instance matching, T=tracking. (cf. Section II)

## I. INTRODUCTION

Robotic manipulation aims to enable robots to reliably execute physical tasks in unstructured environments, requiring generalization to object poses, unseen objects, and scene variations. Imitation learning is a promising approach to achieve this [1], [2], but RGB-based policies still require large amounts of data to handle these variations [2], [3]. This motivates research into intermediate representations for improved generalization [4]–[12].

Keypoints are an attractive option for such a representation: they are compact, encode fine-grained semantics, and can handle intra-category object variations [13], [14]. Previously, obtaining keypoints required training task-specific detectors [13]–[15], limiting their applicability. Advances in vision foundation models [16] now enable extracting keypoints with a single reference annotation and without task-specific finetuning [17], [18]. Several works have leveraged this for keypoint-conditioned imitation learning [9]–[12], demonstrating improved performance over raw RGB representations. However, many open questions remain, including how to best extract these keypoints and in which settings this approach outperforms other representations.

This paper consolidates and builds on the methods proposed in previous works on keypoint imitation learning [9]–[12] and answers key questions about this approach. To this end, we evaluate the generalization capabilities, compare different keypoint extraction methods, and describe failure modes. Our pipeline for keypoint imitation learning consists of three parts. We start by manually specifying 3-6 keypoints per object on a reference image. During training or inference, we use foundation models to extract the corresponding keypoints across variations in poses, objects, and scenes. These keypoints are then lifted to 3D and used as input for a diffusion policy [1], following [9], [10].

We perform over 2000 real-world rollouts[1], spanning two robot platforms and five different tasks. We evaluate both in-distribution performance and generalization to unseen objects and scene variations. Overall, KIL achieves a 75% success rate, compared to 47% for an RGB diffusion baseline, a gap that widens dramatically under scene variations (KIL 70% vs. RGB 10%). KIL performs on par with $S^2$-diffusion [5] (73%), an object-centric baseline that uses estimated depth maps and segmentation masks. We also compare three different keypoint extraction methods (*image matching*, *instance matching*, and *tracking*) but find no significant differences in performance. Finally, we illustrate how current vision foundation models degrade significantly under large variations in object orientations.

*Affiliations anonymized for double-blind submission

[1]kil-manipulation.github.io

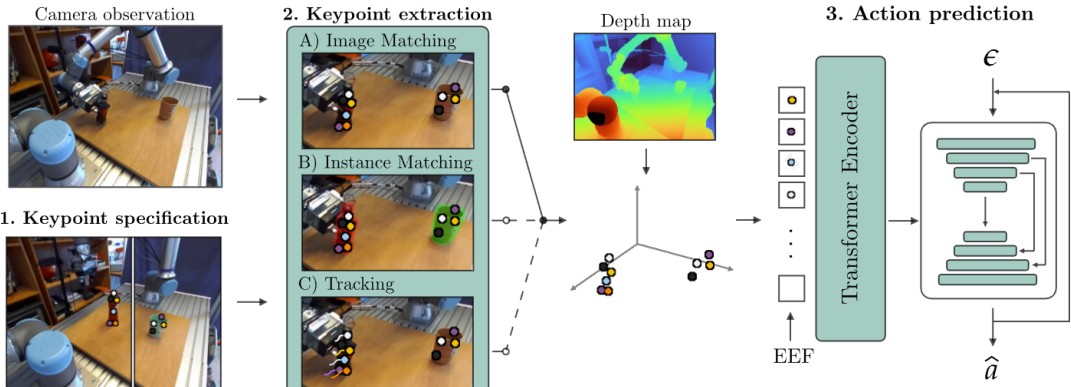

Fig. 2. Overview of our three-stage keypoint imitation learning pipeline. 1) Keypoint references are manually annotated on a reference image for each object category. 2) Keypoints are extracted from each observation with visual foundation models and lifted to 3D. 3) Keypoints are encoded by a transformer and used as input for a diffusion action head that is trained using behavior cloning.

## II. METHOD

Our method for keypoint imitation learning consists of three stages, illustrated in Figure 2. First, a human operator specifies the semantic keypoints of interest by annotating a reference image for each object category (Section II-A). Then, these keypoints are extracted from every observation, for both the demonstrations and during inference (Section II-B). Finally, the resulting sequence of 3D keypoints is encoded by a transformer and used to condition a diffusion policy for action prediction (Section II-C).

### A. Specifying the Keypoint Representation

For each object category the robot interacts with, we manually annotate three to six 2D keypoints on a reference image selected from a single demonstration, similar to [11], [12]. We follow three heuristics when selecting keypoints: minimize occlusions and spread keypoints across the object; place a keypoint near any task-critical contact point (e.g., the heel of a shoe); and maintain a small distance from object borders.

### B. Representation Extraction

Given reference images with annotated keypoints, we extract those keypoints for every observation frame, during both training and inference. Following [9]–[11], we first estimate the 2D position of each keypoint and then lift it to 3D using the intrinsics and extrinsics of a calibrated RGB-D camera.

There are different ways to extract the 2D positions of the semantic keypoints in each frame. In this work, we consider the following three fundamental approaches:

**Image Matching (IM)** Given the reference images and keypoints, we use a visual foundation model to compute a descriptor for each keypoint in the reference image. For each observation frame, we then compute these descriptors for every pixel in the image and use the cosine similarity with the reference descriptor of each keypoint to find the best match, as in [9], [10].

**Instance Matching (IN)** This approach extends image matching by first running an open-vocabulary instance detector [19] (given a prompt for each category) to obtain $N$

instance masks per category, and then finding best match for all keypoints in each mask.

**Tracking (T)** In the third and final approach, we use a dedicated point tracker to extract the keypoints, as in [11], [12], [20]. To initialize the tracker, we use the *instance matching* method to extract keypoints from the initial observation of a demonstration or rollout. Using separate models for tracking and initial matching can boost performance, especially with brief object occlusions.

These three different approaches for extracting keypoints each have different advantages and limitations. We compare them extensively in Section III-F.

### C. Policy Learning

Given the success of diffusion policies for imitation learning [1], we use a diffusion policy as action head to predict robot actions, combined with a transformer encoder for the keypoints, as in [9].

The encoder closely resembles a standard transformer encoder [21], but we use pre-layer normalization, GeLU activations and no dropout. The keypoints and the proprioceptive state are each processed as separate tokens and we use sinusoidal positional embeddings to encode their semantics. The output tokens are mean-pooled into a single vector, following [9]. We use a history of observations as input for the policy, each processed independently in this way. The resulting observation vectors are concatenated and used to condition the diffusion action head, which predicts the next chunk of actions, consisting of the absolute EEF pose and normalized gripper opening. We compare against alternative encoders in Appendix E. During training, we augment the keypoint observations with Gaussian noise and apply random spatial transforms to both observations and actions, following [22]. The impact of these augmentations is evaluated in Appendix F.

## III. EXPERIMENTS

With our experiments, we aim to answer the following questions about KIL:

**RQ-1** Does KIL boost generalization to unseen scenes and objects, compared to alternative methods? (Section III-E)

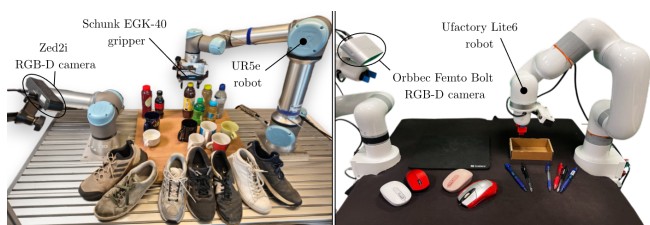

Fig. 3. The 2 robot setups and object sets used for each task (training objects grouped left of the evaluation objects for each task).

**RQ-2** How do the three keypoint extraction methods described in Section II-B perform? (Section III-F)

Next to answering these questions, we illustrate how current foundation models limit the performance of KIL under large object orientation variations in Section III-G.

### A. Hardware Setup

Both hardware setups used in this work are shown in Figure 3, with details provided in Appendix B. A Meta Quest 3 headset running Quest2ROS [23] is used for demonstration collection.

### B. Tasks and Data Collection

To test the capabilities and limitations of KIL, we designed five tasks that require the robot to manipulate different object categories. The choice of objects is largely motivated by prior work [9]–[11], [24]. We collect 50 demonstrations per task, taking about 1 hour per task. This is slightly more than previous works [9]–[11], but provides a practical trade-off between performance and data collection effort. In the demonstrations, we vary the position of the objects significantly during data collection, but, following [9], [10], we limit the variation in orientations to $\pm30°$ around a canonical orientation (cf. Section III-G). As in [9]–[11], we collect demonstrations using two different objects, but without scene variations.

An example of each task and its keypoint representation is shown in Figure 1, while the objects used for data collection and evaluation are shown in Figure 3. Further details on the tasks, the reference keypoint annotations and the distribution of the initial states of the demonstrations are provided in Appendices A,I and H respectively.

### C. Baselines

We compare KIL against two baselines. The first is a standard RGB-based diffusion policy [1]. This baseline uses the same diffusion action head as the keypoint policies, but now conditioned on the output of an image encoder that processes the RGB images (resized to $320 \times 240$). We use the encoder from [1], but do not replace the global average pooling with a spatial softmax. The second baseline is $S^2$-diffusion [5]. This method uses the same image encoder and action head as the RGB baseline, but leverages foundation models to generate a relative, normalized depth image and segmentation mask, which are used as input.

TABLE I. Comparing keypoint extraction methods for KIL against baselines. IM=*IMage matching*, IN=*INstance matching*, T=*Tracking*. Results averaged over five tasks for each generalization axis: in-distribution (in), unseen objects (obj), and scene variations (s).

| Method | Axis Avg. (%) | | | Avg. (%) | CI (%) |
| --- | --- | --- | --- | --- | --- |
| | **in** | **obj** | **s** | | |
| RGB [1] | 74% | 56% | 10% | 47% | [38, 55] |
| $S^2$ [5] | 80% | 74% | 66% | 73% | [66, 80] |
| KIL (IM) | 88% | 68% | 70% | 75% | [68, 82] |
| KIL (IN) | 76% | 70% | 64% | 70% | [62, 77] |
| KIL (T) | 76% | 74% | 74% | 75% | [67, 81] |

### D. Experimental Setup

**Implementation Details** For *image matching (IM)*, we use RADIOv2.5-B [25] as feature extractor, while for *instance matching (IN)*, we use SAM3 [19] to predict instance masks and DIFT [18] with Stable Diffusion 2.1 [26] as featurizer. For *tracking (T)*, we use CoTracker3 [27] (adapted for online tracking in [11]) and SAM3 with DIFT for the initial frame matching. We validate these choices in Appendix D. All images are resized to $640 \times 480$ and we use bilinear interpolation to upscale the output of all feature extractors to the input resolution, as in [9], [24]. Additional details on the implementation, hyperparameters and inference latencies are reported in Appendix G.

**Evaluation protocol** We perform 30 rollouts per task: 10 with in-distribution initial configurations, 10 with unseen objects (using $3-5$ objects per task, shown in Figure 3), and 10 with scene variations, which include distractors and background color changes. We use the same initial states (cf. Appendix J) for the different policies to ensure fairness across policies. Next to providing point estimates of the aggregated success rate or task completion, 95% confidence intervals (CI) are provided to quantify uncertainty and assess statistical significance (details in Appendix C).

### E. Comparing KIL Against Baselines

We compare the performance of KIL against the two baselines, RGB-based diffusion and $S^2$, across all five tasks (**RQ-1**). The results are presented in Table I. KIL performs significantly better than the RGB-based diffusion policy overall, with the RGB policy achieving only 47% success compared to 75% for KIL with *image matching*, the best-performing extraction method (cf. Section III-F). While the RGB policy matches KIL on in-distribution evaluations and generalizes surprisingly well to unseen objects with varying appearance and geometry, it achieves only 10% success on scene variations. $S^2$, by contrast, performs on par with KIL even under scene variations, which noticeably alter $S^2$'s depth maps relative to the demonstrations. Compared to prior work [9]–[11], KIL achieves similar task performances, while our RGB policies perform slightly better, which we attribute to the larger number of training demonstrations.

Imprecise grasping was the most prominent failure mode across all policies: the robot nearly misses the object multiple times until a timeout occurs, or obtains a poor grasp that

TABLE II. Illustration of performance drop when increasing the range of initial orientations for task objects. Evaluated on the *Place Mouse* and *Pick Pen* tasks.

| Method | ±180° | | ±30° | Δ (30→180) |
| | Avg. (NS) | Avg. | Avg. | |
| --- | --- | --- | --- | --- |
| RGB | 70% | 47% | 52% | −10% |
| S² | 80% | 70% | 77% | −9% |
| KIL (IM) | 45% | 35% | 77% | −55% |
| KIL (T) | 48% | 48% | 67% | −28% |

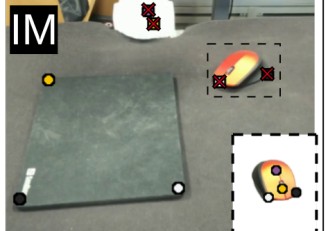 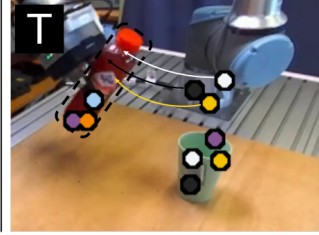

Fig. 4. Failure examples for the keypoint extraction methods. Left (*image matching*): matching assigns keypoints to incorrect parts of the scene rather than the intended object regions; the reference keypoints are shown in the bottom-right. Right (*tracking*): drift causes some keypoints to snap onto a different object.

causes the rollout to fail later on, e.g. by dropping the object in a way that recovery is not possible. For KIL specifically, grasp failures occurred even when the extracted keypoints appeared accurate, suggesting insufficient policy generalization. Even in successful rollouts, KIL occasionally exhibited reduced grasping precision, which we attribute to variance and inaccuracies in the extracted keypoints.

### F. Comparing Keypoint Extraction Methods

We compare the three methods for extracting keypoint representations discussed in Section II-B: *image matching (IM)*, *instance matching (IN)* and *tracking (T)* (**RQ-2**). In Table I, we find that all three methods perform similarly, achieving success rates of 75% (IM), 70% (IN) and 75% (T) on average.

Empirically, *matching*-based approaches for keypoint extraction suffer more from occlusions than the *tracking* approach. However, *tracking* is more prone to drifting over time if objects are occluded for too long, or when the extracted keypoints of the first frame are inaccurate. We illustrate both keypoint failure modes in Figure 4.

### G. Robustness Against Rotations

A surprising limitation of current foundation models used for keypoint features is their poor robustness to large object orientation changes. Combined with their poor calibration, this made us restrict object orientations for all tasks to ±30°. To illustrate this limitation, we create task variants in which object poses span the full ±180°, for both the *Place Mouse* and *Pick Pen* tasks. We also add an orientation constraint to the success criteria: an execution is only considered successful if the object is grasped with an orientation relative to the object that is within ±30° of the canonical orientation used in the demonstrations.

Table II reports the aggregated performance across all initial configurations for both tasks. We also report results excluding scene variations (*Avg. (NS)*). This isolates orientation robustness from scene generalization, where RGB already performs poorly and would otherwise mask the gap. Keypoint-based methods degrade substantially when object orientation spans the full ±180° range and correct gripper approach orientation is required, with KIL (IM) dropping by 55% and KIL (T) by 28%. In contrast, RGB and S² are affected considerably less, dropping by only 10% and 9% respectively.

## IV. DISCUSSION AND CONCLUSION

Similar to [9]–[11], we found KIL to outperform RGB-based policies. However, we found the difference to be most significant under scene variations. An interesting direction for future work is to compare KIL against methods that augment the demonstrations with image generation models [28] or to use pretrained RGB encoders.

Unlike [9], [11], we found our object-centric baseline, S²-diffusion, to perform on par with KIL. This raises the question of why the semantic granularity and compactness of keypoints does not result in superior performance over representations such as masks or depth maps. A possible explanation is that the extraction pipelines (i.e. the foundation models) are more performant for those other representations.

Overall, we observed KIL to be less precise than RGB-based policies, more susceptible to occlusions, and to degrade substantially under large variations in object orientation. These are limitations of the current foundation models used to extract keypoints, rather than the concept of using keypoints for imitation learning. End-to-end finetuning of the extraction pipeline could help to overcome these limitations.

We also identify several limitations of our work: We did not consider the use of wrist cameras, which help to reduce occlusion and can boost generalization for RGB-based policies [29]. We also used depth sensors in this work. Although we found their precision to be sufficient, they have limitations with reflective or transparent objects. The performance of SAM3 [19] for *instance matching* depends on how discriminative the prompt is. For example, in the *Pick Pen* task, we used "paper box" instead of "box" as prompt to improve performance. Such prompt engineering increases the time required for representation specification. Finally, our investigation of KIL's generalization capabilities is limited to 2-shot object generalization and 1-shot scene generalization. A more extensive evaluation of how generalization is influenced by the number of demonstrated variations, as in [3], would provide a more complete picture.

To conclude, this work confirms KIL's potential for improved generalization. More research is needed to overcome its limitations and understand its added value compared to alternative representations and augmentation strategies, and this paper takes a step in that direction.

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

## ACKNOWLEDGMENTS

Funding anonymized for double-blind review.

We used LLMs (GPT5, Claude Sonnet/Opus 4.5) to assist with coding, paper formatting and proofreading.

APPENDIX

## A. Task Descriptions

The success criteria for each task in Section III-B are described below. These criteria are combined with safety measures and a time limit (approx. twice the time it takes to teleoperate the task, excluding inference time) to determine if a rollout is successful.

**Place Shoe**: The goal is to pick a randomly placed shoe by the heel and put it on a tray that is placed at a fixed location. The task is successful if this happens without moving the tray significantly and with the shoe oriented within 90 degrees of the y-axis of the tray.

**Place Mug**: This task is similar to the previous one, but now requires picking a randomly placed, upright mug by the rim and placing it on a tray with a fixed position.

**Pour Bottle**: The robot needs to pick a plastic bottle, move it over a cup, and tilt it as if to pour the contents into the cup. The positions of the bottle and cup are randomized across the workspace. Since both objects are rotationally symmetric around the z-axis, the orientation does not matter for this task. The task is successful if the bottle is tilted beyond horizontal and is held directly above the cup. If the bottle is dropped, the task is terminated.

**Pick Pen**: The goal is to pick up a randomly placed pen and drop it in a cardboard box placed at a fixed location. The task is successful if the pen falls inside or remains on top of the box, without dropping the pen or moving the box significantly.

**Place Mouse**: The goal is to grasp a randomly positioned mouse from the sides and place it on top of a mouse pad placed at a fixed location. The task is successful if the mouse is placed on the computer pad, facing the scene camera with the scroll wheel, without moving the pad significantly or dropping the mouse.

## B. Hardware Setup Details

We use two different setups in this work, as shown in Figure 3. The first setup uses a UR5e Robot with a Schunk EGK-40 gripper and a Zed2i RGB-D camera. We use a custom tactile fingertip [30], to limit the grasping force of the gripper. The second setup consists of a Ufactory lite6 with an integrated gripper and an Orbbec Femto Bolt RGB-D camera. In both setups, the scene cameras are placed as close as possible while having a view of the whole workspace. The cameras are opposing the robot at an inclination of approximately 45 degrees, to limit occlusions by the gripper and self-occlusion of each object in the scene.

Policy and teleoperation actions are generated at 10Hz and then interpolated to 50Hz before sending them to the robot, to ensure smooth tracking.

## C. Statistical Analysis

Next to providing point estimates of the aggregated success rate or task completion, we provide 95% confidence intervals (CI) to quantify uncertainty and assess statistical significance. For binary success results, we assume a binomial distribution, as in [2] and report Clopper-Pearson CIs for the average success rate. For task completion scores, we rely on the central limit theorem ($N = 60$) to assume a normal distribution of the average completion rate, as in [2]. We report the standard Wald CI $\hat{\mu} \pm 1.96\,\hat{\sigma}/\sqrt{N}$.

## D. Comparing Feature Models for Keypoint Matching

To find the most suitable keypoint feature model for each extraction method in Section II-B, we compare three foundation models across different keypoint extraction methods on two tasks: *Place Shoe* and *Place Mug*. The three models are RADIOv2.5-B [25], DIFT [18] with Stable Diffusion 2.1 [26] as featurizer and DINOv3-B [31].

Table III shows that RADIOv2.5-B (85%) performs better than DINOv3-B (73%) and DIFT (62%) for *image matching*. We noticed that, although DIFT produces very accurate matches without occlusions, it is more prone to having incorrect matches, especially when the actual object is partially occluded and with scene variations. Using either *instance matching* or *tracking* reduces the impact of this issue, as can be seen in Table III. Finally, for *instance matching* and *tracking*, we find no significant differences between RADIOv2.5-B and the DIFT models with ensemble sizes of 2 and 8.

Based on these results, we have used RADIOv2.5-B for *image matching*, DIFT$_2$ for *instance matching* and DIFT$_8$ for *tracking* in this paper, unless stated differently.

## E. Comparing Keypoint Encoders

We compare our transformer encoder, described in Section II-B, against two alternatives. The first forgoes any encoder and simply concatenates the keypoints to the proprioceptive state of the diffusion policy, as in [10]. The second uses the same transformer encoder but augments each keypoint's 3D position with the cosine similarity against all keypoint feature descriptors, as in [9].

The results are shown in Table IV. We find that using no encoder performs on par with our transformer encoder, both achieving 85% success rate. Appending cosine similarities to the keypoint 3D positions, however, degrades performance dramatically to 7%. We hypothesize that, in combination with the geometric augmentations used in our pipeline, the network

TABLE III. Comparison of keypoint feature models for the different keypoint extraction methods.

| Extraction Method | Feature Model | place shoe (/30) | place mug (/30) | Avg. (%) | CI (%) |
|---|---|---|---|---|---|
| *Image* | **RADIOv2.5-B** | 25 | 26 | 85% | [73, 93] |
| | DIFT$_2$ | 13 | 24 | 62% | [48, 74] |
| | DINOv3-B | 20 | 24 | 73% | [60, 84] |
| *Instance* | RADIOv2.5-B | 19 | 27 | 77% | [64, 87] |
| | **DIFT$_2$** | 21 | 27 | 80% | [68, 89] |
| *Tracking* | RADIOv2.5-B | 20 | 24 | 73% | [60, 84] |
| | DIFT$_2$ | 23 | 26 | 82% | [70, 90] |
| | **DIFT$_8$** | 21 | 27 | 80% | [68, 89] |

TABLE IV. Encoder Comparison. Evaluated on the *Place Shoe* and *Place Mug* tasks.

| Encoder | Avg. | CI (%) |
|---|---|---|
| None | 85% | [73, 93] |
| **ours** | 85% | [73, 93] |
| + sim | 7% | [2, 16] |
| - aug | 65% | [52, 77] |

TABLE V. Impact of Augmentations on KIL. Evaluated on the *Pick Pen* and *Place Mouse* tasks.

| Augmentations | Avg. CI (%) |
|---|---|
| None | 42% [29, 55] |
| Noise | 80% [68, 89] |
| ST | 70% [57, 81] |
| **Noise + ST** | 77% [64, 87] |

overfits on the similarity scores, which carry spurious information that is easier to memorize than the augmented 3D positions. Removing augmentations substantially recovers performance for the similarity-based variant to 65%, but this is still below the performance we obtained without similarity scores. Beyond this negative interaction with augmentations, we also found the cosine similarity scores to be poorly calibrated across keypoints and scenes. Based on the results from this experiment, we have used the transformer encoder without similarities for all experiments in this paper.

### F. Impact of Augmentations

We study the impact of data augmentations for KIL by selectively enabling the two main augmentations described in Section II-C: adding *Keypoint Noise* and applying a random *Spatial Transform (ST)*. We measure performance on *Pick Pen* and *Place Mouse* using *image matching* as the keypoint extraction method.

Table V reports results. We find that not using any augmentation (42%) hurts performance, but we do not observe meaningful differences between using *KP Noise* (80%), *ST* (70%) or both (77%). For all experiments in this paper, we have used both augmentations.

We also evaluated keypoint dropout augmentation, as in [9], but found it to degrade performance significantly on some tasks, and therefore did not use it in the experiments presented in this paper.

### G. Implementation Details and Hyperparameters

In this section, we provide additional details on the implementation and hyperparameters used for the policies.

Our keypoint transformer encoder consists of 4 layers, using a token size of 128 and an MLP size of 512. The number of learnable parameters is approximately $1M$. We provide a minimal implementation of the encoder architecture used for the keypoint-based policies in Listing 1.

As action head, we use the Unet from [1] and keep most hyperparameters at their default values. The observation history is set to 2. For keypoint-based policies, we halve the number of channels in the Unet, since we did not observe improvements by using a larger action head, in line with [3]. We also use a larger batch size of 128, compared to 32 for the image-based baselines. All policies use a learning rate of $1e^{-4}$ with a cosine scheduler, as in [1]. We train image-based baselines for $100k$ steps, and KIL for $250k$ steps. For all policies, we predict 16 actions and execute the first 8 actions before predicting the next action chunk. The inputs and actions are normalized to (-1,1) using min-max scaling. Unlike the original diffusion policy, we do not use Exponential Moving Averaging (EMA) during training. Table VI summarises the hyperparameters used for training, except for the augmentation hyperparameters, which are reported in Table VII.

The actions are encoded as 10-dimensional vectors, representing the absolute EEF position, absolute rotation (using a 6D representation as in [1]), and normalized gripper state. The rotation is encoded as a six-dimensional vector representing the first two columns of the rotation matrix to make the actions continuous [1]. The proprioceptive state is encoded as a 7-dimensional vector representing the absolute position, absolute orientation using Euler angles, and the normalized gripper state.

Finally, we augment our data by adding zero-mean Gaussian noise to the proprioceptive state and keypoints and applying a random translation and rotation around the $z$-axis to the inputs and actions, as in [22]. We report the hyperparameters for the augmentations in Appendix G and have evaluated the impact of the augmentations on KIL performance in Appendix F.

Table VIII reports the approximate inference latency for each method on an RTX 4090 with torch v2.9.1. We use bf16 precision for foundation model inference and fp32 for policy inference. We do not use torch.compile. The table reports both per-observation latency (preprocessing of each observation at 10Hz, if needed) and the per-inference-step latency (the full cost paid each time the policy is queried for a chunk of actions).

TABLE VI.   Hyperparameters for KIL and baseline policies. All methods use a Diffusion Policy [1] action head.

| Hyperparameter | RGB / S$^2$ | KIL |
|---|---|---|
| Observation encoder | ResNet18 | Transformer |
| Input resolution | $320 \times 240$ | $640 \times 480$ |
| Action Head Dims | [256,512,1024] | [128,256,512] |
| Action Head kernel size | 5 | |
| Noise scheduler | DDIM | |
| Denoising steps | 100 train /14 inference | |
| Action prediction horizon ($T_p$) | 16 | |
| Action horizon ($T_a$) | 8 | |
| Observation horizon ($T_o$) | 2 | |
| Peak LR | $10^{-4}$ | |
| LR schedule | cosine | |
| Optimizer | Adam | AdamW |
| Weight decay | $10^{-6}$ | $10^{-5}$ |
| Batch size | 64 | 128 |
| Training steps | 100K | 250K |
| Augmentations (Table VII) | RandomCrop, Gaussian Noise, ColorJitter | KP Noise, ST, EEF Noise |

TABLE VII.   Augmentation hyperparameters. ST = Spatial Transform; KP = Keypoint; EEF = End-Effector.

| Method | Augmentation | Parameters |
|---|---|---|
| RGB | RandomCrop | $288 \times 216$ |
| | Gaussian Noise | $\sigma = 12.5$ mm |
| | ColorJitter | range$= (0.4, 0.4, 0.4, 0.1)$ |
| S$^2$ | RandomCrop | $288 \times 216$ |
| | Gaussian Noise | $\sigma = 12.5$ px |
| KIL | KP Noise | $\sigma = 1$ mm |
| | ST | $\Delta t = 0.3$ m, $\Delta\theta_z = 0.6$ rad |
| | EEF Noise | $\sigma_{\text{pos}} = 2$ mm, $\sigma_{\text{ori}} = 1$ deg, $\sigma_{\text{grip}} = 0.02$ |

TABLE VIII.   Inference latency (ms) of the different policies used in this work, measured on an NVIDIA RTX 4090 using bf16 precision for foundation model inference. We do not use torch.compile.

| Method | Per observation | Per inference step |
|---|---|---|
| RGB | / | $N_o \times$ ResNet18 + AH : 35 ms |
| S2 | / | $N_o \times$ (DAv2 + gSAM) + AH: 540 ms |
| KIL (IM) | / | $N_o \times$ RADIOv2.5-B + AH: 80 ms |
| KIL (IN) | / | $N_o \times$ (DIFT$_2$ + SAM3) + AH: 570 ms |
| KIL (T) | CoTracker3 : 59 ms | AH : 29ms |

Listing 1.   Minimal implementation of the keypoint encoder

```python
import torch, torch.nn as nn, numpy as np

def sincos_embed(d, n):
    i = np.arange(d // 2) / (d / 2.0)
    e = np.outer(np.arange(n), 1.0 / 10000**i)
    return torch.from_numpy(np.concatenate([np.sin(e), np.cos(e)], axis=1)).float()

class KeypointEncoder(nn.Module):
    def __init__(self, seq_len=9, kp_dim=3, ee_dim=7,d=128, nhead=8, num_layers=4, ff=512):
        super().__init__()
        self.seq_len = seq_len
        self.kp_dim = kp_dim
        self.ee_dim = ee_dim
        self.d = d

        self.kp_emb = nn.Linear(kp_dim, d)
        self.ee_mlp = nn.Sequential(
            nn.Linear(ee_dim, d*2), nn.ReLU(),
            nn.Linear(d*2, d*2), nn.ReLU(),
            nn.Linear(d*2, d), nn.Tanh(),
        )
        layer = nn.TransformerEncoderLayer(d, nhead, ff, dropout=0.0,activation='gelu', batch_first=True,norm_first=True)
        self.encoder = nn.TransformerEncoder(layer, num_layers, norm=None)
        self.register_buffer('pos', sincos_embed(d, seq_len + 1).unsqueeze(0))

    def forward(self, kp, ee):
        B, T = kp.shape[:2]
        kp_tokens = self.kp_emb(kp.reshape(B*T, self.seq_len, self.kp_dim)) + self.pos[:, :self.seq_len]
        ee_token = self.ee_mlp(ee.reshape(B*T, self.ee_dim)).unsqueeze(1) + self.pos[:, self.seq_len:]
        tokens = torch.cat([kp_tokens, ee_token], dim=1)
        tokens = self.encoder(tokens)
        return tokens.mean(dim=1).reshape(B, T, -1)
```

## H. Initial Scene Configurations of Demonstrations

Figure 5 shows an overlay of the initial frames of the demonstrations for each task, illustrating the distribution of initial scene configurations used during data collection. As described in Section III-B, we vary the position of objects significantly across the workspace, but limit orientation variation to $\pm 30°$ around a canonical orientation. Demonstrations are collected using two different objects per task, without scene variations.

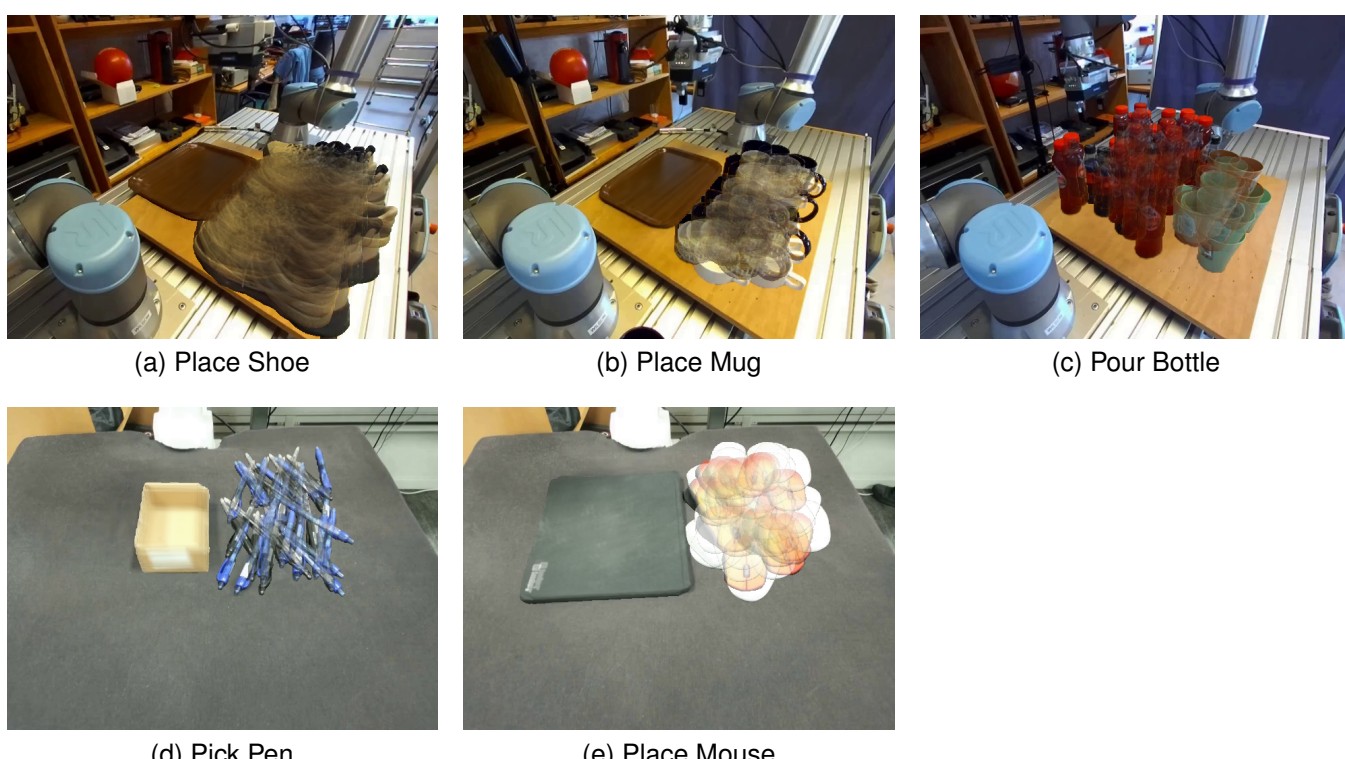

(a) Place Shoe      (b) Place Mug      (c) Pour Bottle

(d) Pick Pen      (e) Place Mouse

Fig. 5. Scene configuration distribution of the demonstrations for each task.

## I. Keypoint Representation Reference images

Figure 6 shows the keypoint reference images used for each task, as described in Section II-A. Each image shows the reference frame with the annotated keypoints and text prompt used for instance detection (cf. Section II-B).

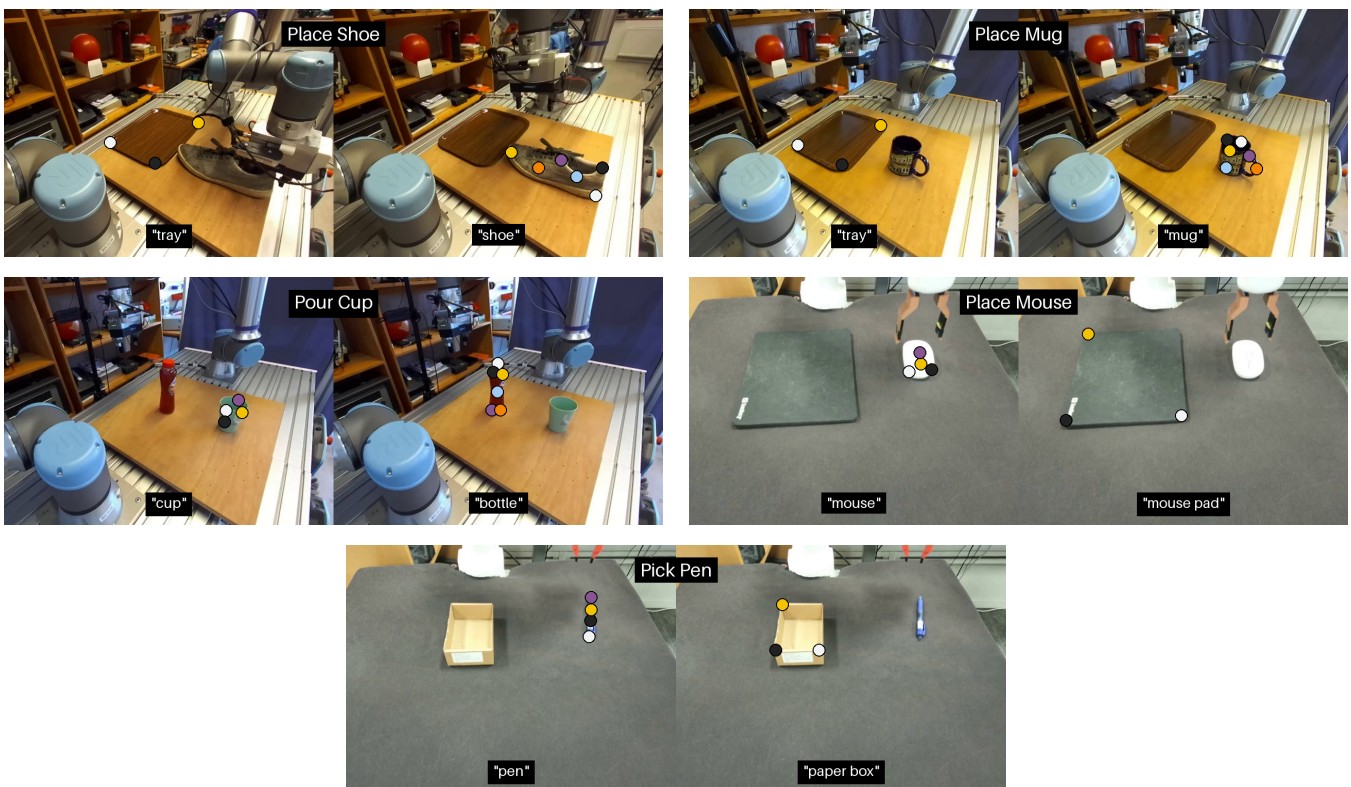

Fig. 6. Keypoint reference images for each task, including text prompts for each object.

## J. Initial Scene Configurations for Evaluations

Figure 7 shows the initial scene configurations used for evaluation. For each task, we measure in-distribution performance, generalization to unseen objects and generalization to scene variations. We use the same 10 initial scene configurations for all policies to ensure fairness, by overlaying reference images on the current observation and manually positioning all objects accordingly. Additionally, we control lighting conditions and background elements to limit distribution shifts between evaluations.

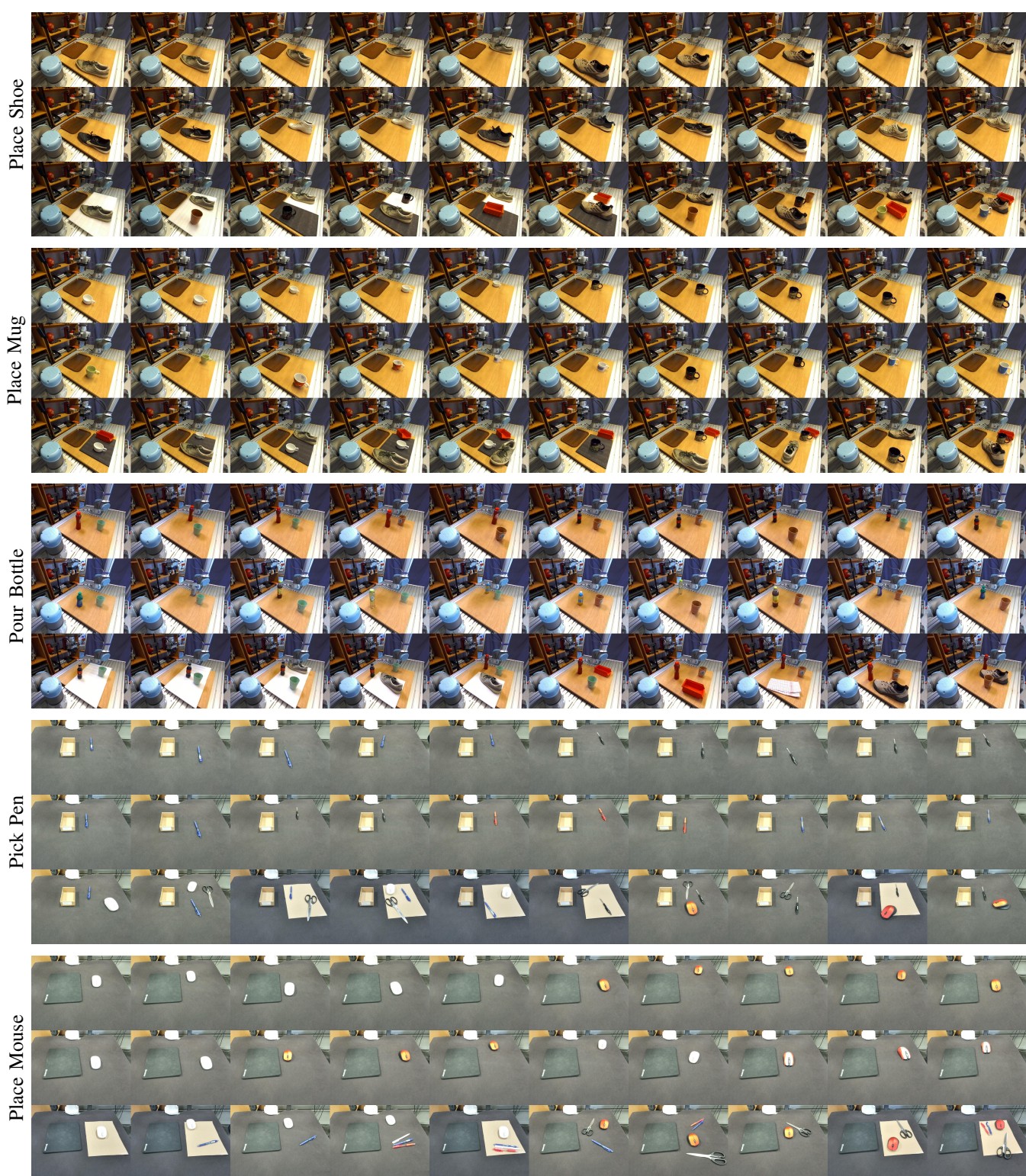

Fig. 7. Initial scene configurations used for evaluation. For each task, we use 10 in-distribution configurations (first row), 10 unseen object configurations (second row) and 10 scene variation configurations (third row).