# OpenReview forum: "On the Generalization Capabilities, Design Choices and Limitations of Keypoint Imitation Learning"
_IEEE.org/ICRA/2026/Workshop/Manipulation_Robustness — ICRA 2026_

### Official Review · Reviewer_Zvdx · 2026-05-11
**The paper provides a detailed analysis of keypoint imitation learning (KIL), demonstrating KIL as a competitive method for achieving object, spatial, and scene generalization in low-data training regimes.**

**Rating:** 7
**Confidence:** 4

**Review:**

### Strengths:
- The work provides insights across all stages of the KIL pipeline, evaluating keypoint encoders (raw keypoints vs transformer encoder), and keypoint data augmentation strategies.
- The authors analyze three frameworks for extracting keypoints from training episodes (Image Matching vs Instance Matching vs Tracking), discussing the successes and failures of each.
- The paper presents a robust evaluation of KIL: comparing against two baselines across five tasks and two hardware platforms, testing the performance of KIL on in-distribution objects, out-of-distribution objects, and robustness against environment perturbations.

### Weakness:
- In Appendix G, the authors report that the RGB baseline is trained with a smaller batch size (32 vs 128) and for fewer steps (100k vs 250k), resulting in ~10$\times$ fewer training samples (3.2M vs 32M) than KIL. For a more controlled comparison it would be helpful to train the RGB baseline for the same number of steps or until convergence. Additionally although acknowledged in the future work, the paper would benefit significantly from a comparison between KIL and a RGB baseline that uses more modern, pretrained image encoders (e.g. VIT-B/16 or SigLIP)
- The paper would benefit from clearer acronym usage (e.g. changing the abbreviation for “in-distribution” from “IN” to “ID” and defining Table 2’s acronyms in its caption)

---

### Official Review · Reviewer_E81W · 2026-05-16
**Systematic empirical study of KIL with insightful negative findings. The comparison between KIL and S2-Diffusion and the cross-representation synthesis deserve deeper analysis.**

**Rating:** 7
**Confidence:** 4

**Review:**

This paper provides a systematic empirical study of keypoint imitation learning (KIL), consolidating design choices from prior work into a clear three-stage pipeline and focusing on the tradeoffs between keypoint and alternative representations under scene variation, object variation, and robustness to object orientation. The authors compare KIL against RGB and $S^2$-Diffusion baselines across five real-world tasks on two robot setups, with evaluations of three keypoint extraction methods and a characterization of a clear failure mode of certain KIL variants under large orientation changes.


Strengths:
- The paper is well written and guides the reader step by step through its setup and the comparison between baselines, with clear and insightful framing.
- The paper shows that $S^2$-Diffusion performs on par with KIL overall (73\% vs. 75\%) and remains competitive under scene variations, which challenges the prior framing that semantic-keypoint compactness should dominate object-centric alternatives.
- The evaluation spans five real-world tasks across two robot platforms with consistent configurations across policies, providing a rigorous basis for understanding the systematic differences between KIL and other diffusion-based methods.


Improvements:
- In Table I, $S^2$-Diffusion shows consistent performance across conditions: 80\% in-distribution, 74\% on unseen objects, and 66\% on scene variations, compared to KIL (IM) at 88/68/70\% and KIL (T) at 76/74/74\%. The overall averages closely match (73\% vs. 75\%), with  $S^2$-Diffusion showing only a modest drop under scene variations. The authors flag this parity as surprising in Section IV and offer one speculative sentence about extraction-pipeline maturity, but do not investigate where the actual bottleneck lies. A degraded-input study or per-rollout failure-mode comparison with KIL would strengthen this finding.
- The reduced grasping precision attributed to keypoint variance may also reflect the single oblique third-person camera setup, where limited pixel resolution per object propagates into 3D grasp-pose error. A wrist-mounted or closer view would help isolate this from the representation itself.
- The paper's scope is framed around design choices and limitations of KIL, but the discussion of how the different keypoint extraction methods fail under scene variation, object variation, and orientation changes remains limited. The findings point to a more nuanced picture in which keypoints, depth-plus-mask, and pretrained RGB features may each be preferable in different regimes, yet the paper stops short of articulating this. A clearer statement of which task properties favor which representation, and of which extraction method works best under which conditions, would strengthen the consolidation contribution and provide sharper insight into the trade-offs across different KIL methods (IM, T, and IN).

---

### Decision · Program_Chairs · 2026-05-21

Accept